# Looking into the Crystal Ball—How Automated Fast-Time Simulation Can Support Probabilistic Airport Management Decisions

Oliver Pohling *, Sebastian Schier-Morgenthal and Sandro Lorenz

German Aerospace Center (DLR), Institute of Flight Guidance, Lilienthalplatz 7, 38108 Braunschweig, Germany; sebastian.schier@dlr.de (S.S.-M.); sandro.lorenz@dlr.de (S.L.)
* Correspondence: oliver.pohling@dlr.de

**Abstract:** Airport management plays a key role in the air traffic system. Introducing resources at the right time can minimize the effects of disruptions, reduce delays, and save costs as well as optimize the carbon footprint of the airport. Efficient decision-making is a challenge due to the uncertainty of the upcoming events and the results of the applied countermeasures. So-called 'what-if' systems are under research to support the decision-makers. These systems consist of a user interface, a case management system, and a prediction engine. Within this paper, we evaluate different types of prediction engines (flow, event, and motion models) that can be used for airport management what-if systems by comparing them in terms of accuracy and calculation speed. Hence, two different operational situations are examined to evaluate the performance of the prediction engines. The comparison shows that accuracy and calculation speed are opposed. The flow model has the lowest accuracy but the shortest calculation time and the motion model has the highest accuracy but the longest calculation time. The event model lies between the other two models. The acceptable accuracy of a prediction tool is strongly dependent on the respective airport, whereas the calculation time is strongly dependent on the available decision time. Regarding airport management, this means that the selection of a prediction engine has to be made in dependence of the airport and the decision processes. The results show the advantages and disadvantages of each prediction engine and provide a first quantification by which a selection for what-if systems can happen.

**Keywords:** fast-time simulation; airport management; what-if





## 1. Introduction

Airport management has to deal with a large variety of events. In most cases, these events own a degree of uncertainty regarding their occurrence and impact in advance. Weather situations in particular challenge the airport management by their degree of uncertainty. For instance, whether a cold front system generates freezing fog or snow depends on small changes in humidity and temperature [1]. Minor weather deteriorations can have a negative impact, and they may lead to delays and flight cancellations [2]. To maintain continuous airport operations in these circumstances, a joint approach evaluating each possible development with all stakeholders is necessary, but time consuming.

A joint approach is necessary since each stakeholder plans separately without having an overview of the consequences that each action has for other stakeholders and vice versa. Total airport management addresses this problem, as it enables the development of joint actions to react to future events that have negative impacts on the airport's performance [3]. Within this process, what-if tools are possible instruments that facilitate decision-making [4]. What-if tools forecast the outcomes of events and chosen actions. As stakeholders will rely on these forecasts to make their decisions, the determination of the outcomes has to be very accurate on one side and easily comprehensible on the other [3]. Since the manual calculation of potential scenarios is time consuming, stakeholders are not able to evaluate

all scenarios. Fast-time simulation is a forecast tool that may be used to overcome this problem because of its time advantage.

The use of simulation models represents an improvement compared with the status quo. As a result, this leads to the research question of which simulation model is best placed to act as a prediction engine for what-if systems. The present paper addresses this challenge. Regarding the demand for a fast, accurate, and comprehensible forecast, an appropriate model needs to be chosen. As such, we compare different models in predicting the impacts of a snow situation at Oslo airport.

The remainder of this paper is structured as follows. The Section 2 gives a brief overview of existing prediction models and how they can be categorized. The Section 3 examines the approach to the comparison of different prediction models. It comprises the simulation scenario and setup, the definition of key performance indicators (KPIs), and the validation method. The Section 4 presents the simulation results of each considered model. In the Section 5, the results are discussed and an assessment of the prediction models is performed. Our conclusions are drawn in the Section 6.

## 2. Existing Work

This section will provide an overview of existing prediction models for airport operations. In the Section 2.1, basic airport model categories are derived. For each category, a representative is selected and described in the Section 2.2 of this section. Finally, the Section 2.3 provides a summary of previous experiences with different kinds of airport models in what-if applications.

### 2.1. Airport Model Categories

The research offers a broad range of airport models that have been designed to fast-time simulate and predict various aspects of airport operations. A structured literature analysis (conducted in September 2021) using Google Scholar, Crossref, Scopus, and Science Direct with the key words "airport simulation model" revealed more than 1000 publications. The results cover all kinds of airport models, from specific models for distinct aspects and airports to comprehensive models for general research questions. It was assumed that additional key words (e.g., "airport prediction model", "airport resource model", "air traffic simulation") would enlarge the resulting set.

In contrast to the large number of results for the common search of "airport simulation model", only a small number of results were retrieved once the what-if focus was added to the search phrase ("what-if airport airside simulation model"). Initially, 330 results were delivered by the search engines. A manual analysis of titles and abstracts was applied to filter out, for instance, airport capacity studies, simulations that focused only on one distinct airport resource (e.g., runways, ground handling), or studies that focused on landside processes only. In the end, 29 publications dealing with what-if simulation models for airports were identified as providing valid models. Upon analyzing these publications, it became clear that none of them discuss the question of which simulation model is best placed to act as a prediction engine for what-if systems.

As several general airport simulation models exist, but a comparison for what-if systems has not been conducted, simulation models of different types were selected for evaluation. Therefore, we categorized the simulation models. The categorization of simulation models in general is a challenge due to the different characteristics of simulations (e.g., level of detail, representation of time, simulation functionality) [5]. A categorization regarding the representation and calculation of basic data as required for this study was derived by a distinction between simulation time and object representation [6]. Simulation time and objects can either be discrete or continuous. Table 1 shows the resulting simulation categories that were derived by an application to the airport context. In this context, the simulation objects are flights.

**Table 1.** Types of airport simulation models (adapted from Fishwick [6]).

| | | Flights (Simulation Objects) | |
| --- | --- | --- | --- |
| | | **Discrete** | **Continuous** |
| **Time** | discrete | Flow model | Event model |
| | continuous | State model | Motion model |

In general, the flow model represents the air traffic as streams or flows and discretizes the flights as active (part of the stream) or not active (not part of the stream). Multiple flows (e.g., arrival/departure flows) are necessary to model the airport. In comparison with the flow model, a flight in the state model can have more states than active and not active (e.g., landing and in-block). Therefore, the simulation time can be continuous and state changes throughout the time can be observed. The event model is capable of simulating flights with all their continuous attributes (e.g., altitude, speed, etc.). Events, which happen at discrete points in time, have an impact on these attributes (e.g., a blocked runway leads to an aircraft taxi speed equal to zero). In contrast to the event model, the motion model consists of multiple equations by which the movement of each flight can be calculated at any given point in time.

*2.2. Existing Airport Model Implementations*

The above-defined categories were represented by one model implementation each. To later discuss the differences between the simulation models, it was necessary that either the documentation on the model algorithms or the source code be available for further analysis. Moreover, the models needed to be adaptable in such a way that each model would be capable of simulating the same set of traffic flows and occurring events.

As a flow model, a prototype from the German Aerospace Center (DLR) specifically designed for demand–capacity balancing of total airport management (cf. [7]) was selected. This prototype has not been published yet, and as such the general functionality will be explained here. The prototype derives the demand for the arrival runway, the airport stands, the ground handling, and the departure runway from the flight plans on an hourly basis. The hourly capacity of each resource must be configured before running the model. Out of a comparison between demand and capacity, the flow is calculated. If the demand exceeds the capacity, the so-called 'overdemand' is delayed until the next hour.

A sufficient state model for what-if applications could not be retrieved. Although multiple state models for flight description do exist (e.g., [8,9]), none have been adapted for comparable operations with the other models or had sufficient documentation for the comparison. As such, the state model was left for future work.

The event model was derived from the DLR's airport management simulation platform (cf. [10]). It was published as the "milestone simulation" and is described in [11]. This simulation picks up the 'milestone' concept of airport collaborative decision-making. Each flight is a series of processes where all processes end with a milestone. Once a flight requires an airport resource (e.g., a runway, a stand, ground handling staff), it sends a request and receives a clearance as soon as the resource is available. Milestones, requests, and clearances are designed as events that happen at a certain point in time. The simulation calculates the events, stores them in an event queue, and jumps from one event to the next event.

As a representative for the motion model, the fast-time simulator "Air Traffic Optimizer" (AirTOP) was chosen in this study. AirTOP was developed by Airtopsoft and allows for the assessment of air traffic operations at the airport, inside the terminal maneuvering area (TMA), or within the en route segment. Thus, it is called a gate-to-gate simulation. AirTOP uses a rule-based approach to set up the simulation, including separation standards, rules for conflict resolution, and runway dependencies. The modeling of multiple agents and their tasks, for example radar and airport controllers, is a significant element of AirTOP because the agents pass on the instructions to the aircraft to comply with the prescribed rules [12]. In regard to the focus on airport operations in this study, the fast-time simulation

tool enables a detailed modeling of the ground layout and the ground operations. This includes taxiways, aircraft stands plus their allocation, runway entries and exits [12], and de-icing procedures. As a result, the movements on the apron are simulated and the aircraft take other moving aircraft into account due to the rule-based modeling. This simulation of ground operations allows for, e.g., the estimation of taxiing durations and the evaluation of resulting ground delays. A useful feature is the integrated reporting function in AirTOP [12] because it allows us to comprehend the simulation result, which is one of the key elements of a what-if tool. As an example, reports are available regarding various delays, runway throughput, used aircraft stands, and burned fuel. In conclusion, any property of any object can be reported, which enables the user to obtain the desired information.

*2.3. Former Approaches*

This subsection provides a description of existing research works regarding the usage of airport simulation for what-if purposes.

Timar et al. [13] developed a prototype what-if tool in regard to the prediction of traffic performance at airports. The aim of this tool is to minimize the effects of demand peaks that lead to imbalances at the airport if they exceed the airport's capacity. The tool utilizes flight plan information, an airport and airspace model, and flow management regulations as input data. In terms of the forecast tool, the used motion model is based on a node–link graph displaying the airport layout, where the nodes represent aircraft stands, runways, and airside fixes and the links map the taxiways and airside routes by connecting the nodes. Furthermore, the nodes have a service time and therefore only allow for the passing of a specified number of movements. Besides the motion model, the prototype what-if tool consists of a departure management emulation to test various flow management actions for the departing traffic. The different departure management actions can then be compared because the fast-time simulation forecasts the traffic with and without the chosen actions. Finally, the results are evaluated by means of KPIs [13].

Another approach was given by Zografos et al. [14] with the decision support tool "Supporting Platform for Airport Decision Making and Efficiency Analysis Decision Support System" (SPADE DSS). The system enables the evaluation of effects on the airport's performance that occur due to adjustments carried out by the user. These adjustments are divided into three categories: infrastructural changes (e.g., an additional runway); operational changes (e.g., new flight procedures); and traffic changes, such as flight diversions. In general, the approach is applied with use cases, which are pre-defined simulation configurations. SPADE DSS relies on fast-time simulation as part of its control component. Depending on the use case, different tools are selected that are based on microscopic or macroscopic models. These tools are used for the execution of the use cases and to evaluate the airport's performance, e.g., airport capacity, level of service, and delays. The design of this what-if tool supports the decision-making on a strategical or tactical level.

Günther et al. [15] discussed the idea of coupling a pre-tactical planning system with a fast-time simulation to support the DLR's concept of "Performance-Based Airport Management (PBAM)", providing certain KPIs as drivers for airport steering and control. For this purpose, a predefined data set, including target times at the runway (planned by the pre-tactical planning tool "Total Operations Planner"), estimated times, and actual times (derived from an AirTOP simulation), was exchanged. The implementation was realized by running cycles of the following steps: the fast-time simulation stops the simulation at a defined time; the planning tool receives estimated and actual times plus flight diversions at the time the simulation freezes to update its own airport operations plan data; the planning tool calculates times for a defined forecast time horizon; and the fast-time simulation sets time constraints depending on this plan and resumes the simulation. The results showed that both systems were able to properly handle the data computed by the other tool. Furthermore, appropriate KPIs in terms of delay could be determined.

## 3. Method

In this section, the approach to the comparison of the airport simulation models is described. The focus is on the simulation setup and process, the parameters to be researched, and the validation method for the different forecasts. As an initial step, the general validation design is outlined, followed by the traffic scenario selection and the transfer to the individual simulation setups. This section is concluded by a selection of comparison parameters.

### 3.1. Validation Design

The general research question of which airport simulation model is most suitable for a what-if tool was answered by a comparison among flow, event, and motion models. A comparison of simulation models can be performed regarding multiple aspects. In the case of a what-if tool, two general factors are of major importance. On the one hand, the prediction must be precise enough to ensure effective actions (e.g., flight cancellations, calls for additional staff) from the airport management. On the other hand, the prediction must be available in a short amount of time to allow for the conduction of multiple scenarios and enable a structured decision-making process among the stakeholders.

The comparison in terms of precision and calculation speed was performed as an experiment with four conditions. Each of the three simulation models was validated within one condition. Additionally, a baseline was necessary as a fourth condition to evaluate the precision of the models. The baseline contained the actual flight times while the simulation models were challenged by operational flight plan data.

As a what-if tool has the potential to provide support in a broad range of operational situations, a large set of possible operational scenarios must be considered within a comparison. This study concentrated on one nominal scenario without weather limitations and on a second non-nominal scenario where weather is a major restriction to flight operations. In evaluating these two situations, an insight into the performance of the simulation models will be provided. An overview of the validation design is provided in Figure 1.

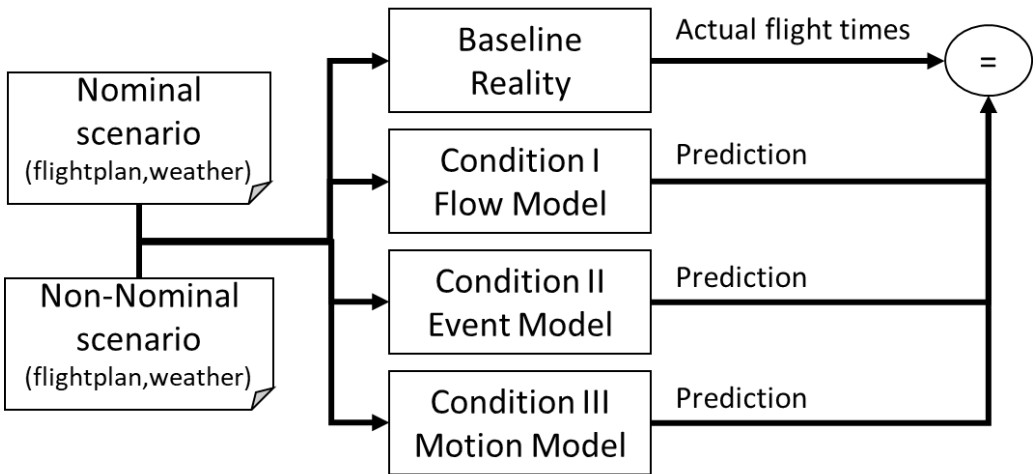

**Figure 1.** Validation design.

Each simulation model can be viewed as part of a so-called 'input–output transformation'. A general representation of such a transformation can be seen in Figure 2. As its name implies, input data are added to a system where the data will be processed. Afterwards, the system returns the data as a transformed output [16].

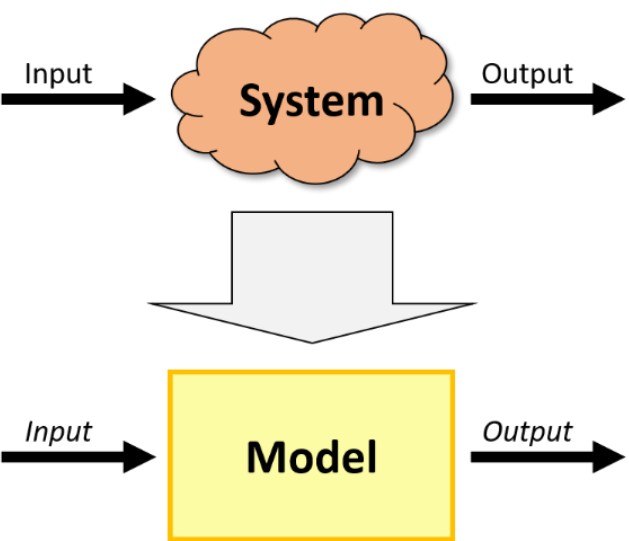

**Figure 2.** Input–output transformation (based on [16]).

The aim of our approach is to validate whether the models represent the real system properly, especially in terms of predicting the outcomes of chosen actions, which means the future behavior. Therefore, a comparison with historical data (actual flight times) is necessary. We are aware that our approach has a limitation because the actual processes and actions inside the system, e.g., stakeholder decisions, are unknown to us. Therefore, we had to deduce possible actions that were applied in reality.

### 3.2. Scenario Selection and Analysis

The comparison was based on the airport data from Oslo Gardermoen as shown in Figure 3. Oslo operates with two independent runways. The terminal is located between these runways. Oslo has been subject to multiple airport management simulations conducted by the German Aerospace Center [17,18]. Therefore, knowledge about the operational procedures is available. Moreover, Oslo will also be subject to a human-in-the-loop simulation that assesses the benefits of what-if from a user perspective [3]. Using the same data set as in this human-in-the-loop simulation enables a comparison between the user demand and the potential raised within this study in future work.

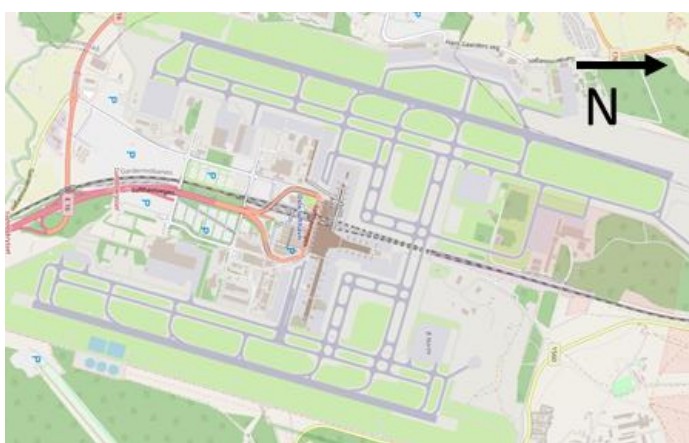

**Figure 3.** Oslo Gardermoen airport (with north located to the left) [19].

In addition to the airport selection, traffic and weather conditions need to be defined. As stated above, one nominal case and one non-nominal case were evaluated. Judging the precision and calculation speed under these conditions requires real-world complexity

within the data. Therefore, two days of operation at Oslo airport were chosen. The flight plan data and the weather data of these days served as the input for the simulation models. The actual flight times provided the baseline results to be compared with the results of each simulation model.

### 3.2.1. Nominal Scenario

As a nominal day of operation, 15 July 2020 was chosen. The flight plan includes 339 flights and was provided by EUROCONTROL's Demand Data Repository (DDR2), which has been used as a source for various studies (e.g., [20–22]). DDR2 provides SO6 files that are distinguished in two models: M1, which contains trajectories computed by the last filed flight plan, and M3, which is a modified M1 model updated with radar information [23]. M3 is regarded by the network manager systems as flown trajectories.

The Meteorological Aviation Routine Weather Report (METAR) [24] describes the weather on 15 July 2020 as stable with air pressures between 1012 hPA and 1009 hPA, temperatures in the range from 12 °C to 19 °C, and temporary showers in the evening. No special weather events (e.g., fog or thunderstorms) were reported. Icing conditions were not present due to the temperature. Therefore, it was concluded that flight operations were not affected by the weather.

An analysis of the trajectory data reveals runway 01L as the active runway throughout the entire day. Although Oslo airport owns two runways (01L and 01R) [25], only runway 01L was utilized due to the impact of COVID-19 on the global air traffic and the resulting low traffic volume [26].

### 3.2.2. Non-Nominal Scenario

For the non-nominal scenario, 20 October 2020 was chosen. The flight plan data were provided by DDR2 and, besides a similar traffic volume (338 flights), it was taken into consideration that the utilized runway direction (01L) is identical to the nominal scenario. This ensured comparability between the two scenarios.

According to METAR data, the temperatures were in the range from 0 °C to 2 °C and rain showers with a temporary transition to snowfall occurred the entire day. A phase of significant snowfall was reported from 01:00 a.m. to 11:00 a.m. UTC. These weather conditions are a common challenge for Oslo airport in the winter season [1].

Based on the weather conditions and the traffic data, operational limitations can be derived. Snowfall affects the runway system by covering the surface and reducing friction. Therefore, snow removal is an important action to take to ensure safe runway operations. We analyzed the timestamps of the runway movements to find gaps greater than 15 min between two consecutive movements that could indicate a temporary runway closure for snow removal during the period of snowfall (cf., Figure 4).

Additionally, the real flown trajectories (arrival and departure flights) of the respective days were considered in order to precisely determine the timestamps since the gaps between two movements are not the sole indicator of snow removal. Gaps greater than 15 min were excluded if they occurred during periods of low traffic demand, e.g., from 00:00 a.m. to 05:00 a.m. UTC. Afterwards, four striking timestamps were apparent on 20 October 2020, namely at 05:40 a.m., 07:10 a.m., 08:30 a.m., and 10:25 a.m. UTC, where each gap was about 20 min between two movements. In our view, it can be concluded that these four gaps resulted from the removal of snow from the runway because of their consistent interval (in each case, 90 min) and the nearly identical duration of the gap. Furthermore, the trajectories (cf., Figure 5a) revealed that arrival flights were subject to delaying maneuvers by the air traffic control. Aircraft flew holding patterns at each of the four timestamps and were affected by path-stretching on the point merge leg. This suggests that the runway was closed, and the aircraft were delayed for this reason. Figure 4 shows that there are two gaps that are longer and one gap that is slightly shorter than 15 min between 08:00 a.m. and 09:00 a.m. UTC on 15 July 2020. These gaps are not as consistent and do not have an identical duration like the ones on 20 October 2020. Beyond that, Figure 5b shows

the real flown trajectories of arrival flights on 15 July 2020, and it points out that arrival flights were not subject to delaying maneuvers such as holding patterns or path-stretching. Therefore, we can assume that these gaps occurred due to a period of low traffic demand. As a conclusion of our conducted data analysis, we determined a duration of 20 min for the snow removal action.

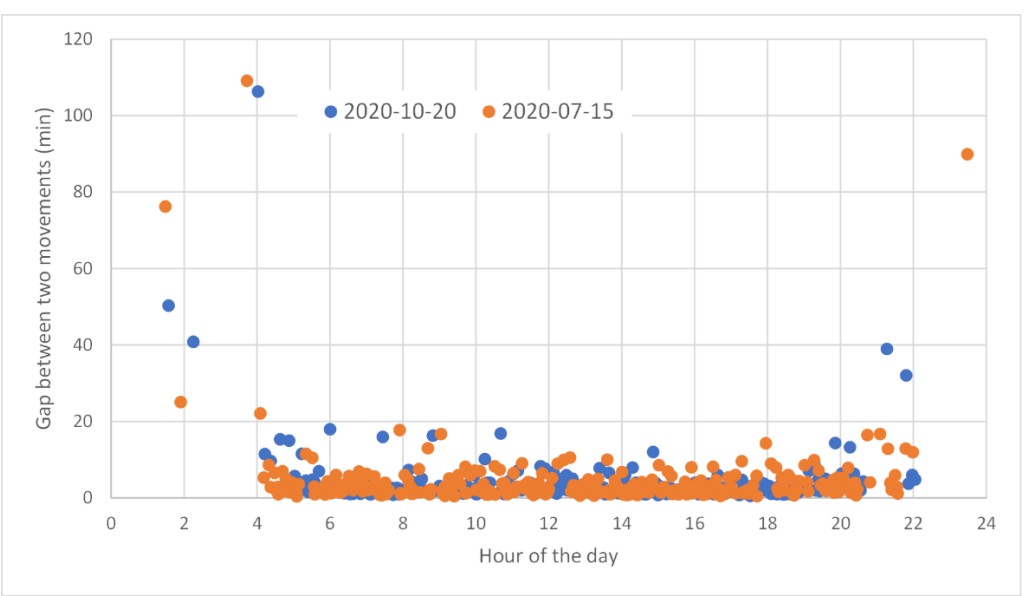

**Figure 4.** Gaps between consecutive movements on runway 01L.

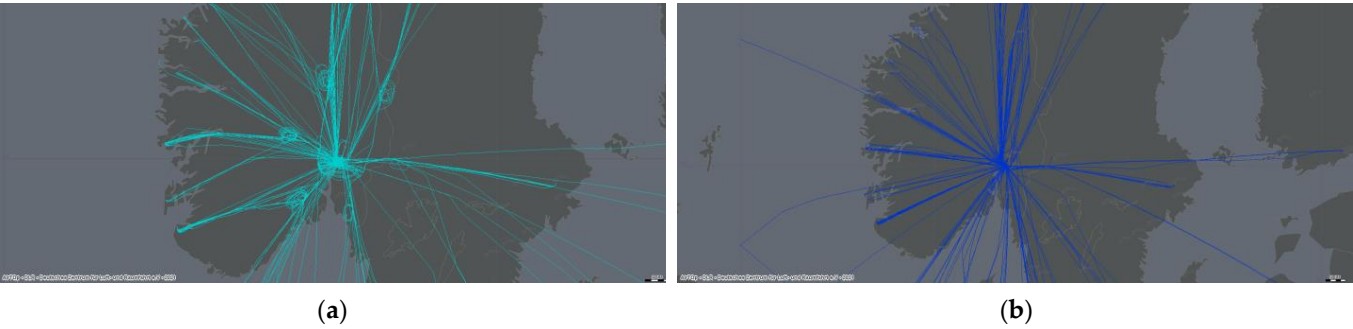

**Figure 5.** Real flown trajectories of arrival flights to Oslo airport: (**a**) from 20 October 2020; (**b**) from 15 July 2020.

Besides snow removal, winter conditions can limit airport operations by additional de-icing of the aircraft. De-icing is necessary because the aircraft surface has to be clean of any contamination, such as ice, slush, or snow, in order to ensure controllability and unimpaired aerodynamic performance [27]. This is the case when significant snowfall occurs as happened between 01:00 a.m. and 11:00 a.m. UTC on 20 October 2020. According to International Civil Aviation Organization (ICAO) Doc 9640, ice or frost can form on the aircraft surface even at temperatures above the freezing point (e.g., after 11:00 a.m.) [28]. The effects on each individual flight could not be derived from the provided data. Therefore, it was assumed that a so-called 'one-step' de-icing procedure [28] for all departing aircraft from 01:00 a.m. to 11:00 a.m. was applied. The duration of this one-step de-icing procedure can be determined from the required quantity of de-icing fluid per aircraft type and the fluid application rate of the de-icing vehicle. Table 2 lists the recommended amounts of de-icing fluid per aircraft type in liters [29].

**Table 2.** Aircraft-type-specific amount of de-icing fluid (data from [29]).

| Airbus A300 | Airbus 310 | Airbus A320 | Airbus A330/340 | Airbus A380 | Boeing 736/7/8 | Boeing 747-8 | Boeing 777ER |
|---|---|---|---|---|---|---|---|
| 363 l | 370 l | 230 l | 580 l | 1130 l | 230 l | 875 l | 705 l |

An example of a typical de-icing vehicle is the Vestergaard "Elephant BETA-15". It is capable of de-icing aircraft up to an Airbus A380. The fluid application rate ranges from 20 L/min to 240 L/min [30]. This results in two duration values, which is why the mean of both values was taken as the standard duration for the fluid application. Because of an information shortage regarding the technical functionality of the de-icing vehicle, we took the mean of the minimum and maximum duration values instead of calculating a duration on the basis of the average fluid application rate. Table 3 shows the results of this calculation. The duration resulting from using the maximum fluid application rate is labeled as "Min", whereas the duration resulting from using the minimum fluid application rate is labeled as "Max".

**Table 3.** Aircraft-type-specific duration for fluid application (in minutes).

|  | Airbus A300 | Airbus 310 | Airbus A320 | Airbus A330/340 | Airbus A380 | Boeing 736/7/8 | Boeing 747-8 | Boeing 777ER |
|---|---|---|---|---|---|---|---|---|
| Code letter | D | D | C | E | F | C | F | E |
| Min | 1.5 | 1.5 | 1 | 2.4 | 4.7 | 1 | 3.6 | 2.9 |
| Max | 18.2 | 18.5 | 11.5 | 29 | 56.5 | 11.5 | 43.8 | 35.3 |
| Mean | 9.85 | 10 | 6.25 | 15.7 | 30.6 | 6.25 | 23.7 | 19.1 |

To include these values in the simulation models, a generalization was made. The aircraft-type-specific durations for fluid application were merged into their respective aircraft size category according to the code letter of the ICAO aerodrome reference code (A to F). If there were discrepancies between the aircraft types of the same code letter, the greater value was used. In addition, the duration for aircraft of code letters A and B had to be assumed because of a lack of data. The aircraft size category-specific durations for fluid application, which were considered in the simulation model, are presented in Table 4.

**Table 4.** Aircraft size category-specific duration for fluid application (in minutes).

| A | B | C | D | E | F |
|---|---|---|---|---|---|
| 5 | 5.5 | 6.25 | 10 | 19.1 | 30.6 |

*3.3. Simulation Configuration*

3.3.1. Flow Model

The DLR flow model requires the capacity to be expressed as the number of maximum flights per hour an airport resource is able to handle (e.g., 30 flights per hour for the arrival runway). The capacity was derived from the Aeronautical Information Publication (AIP) [25] and the information published by Oslo airport within EUROCONTROL's airport corner [26]. Besides capacity, the demand, which means the number of arriving and departing flights per hour, is mandatory as an input for the flow model. The number of flights was derived from an analysis of the flight plan data. Figure 6 shows an extract and visualization of the input data.

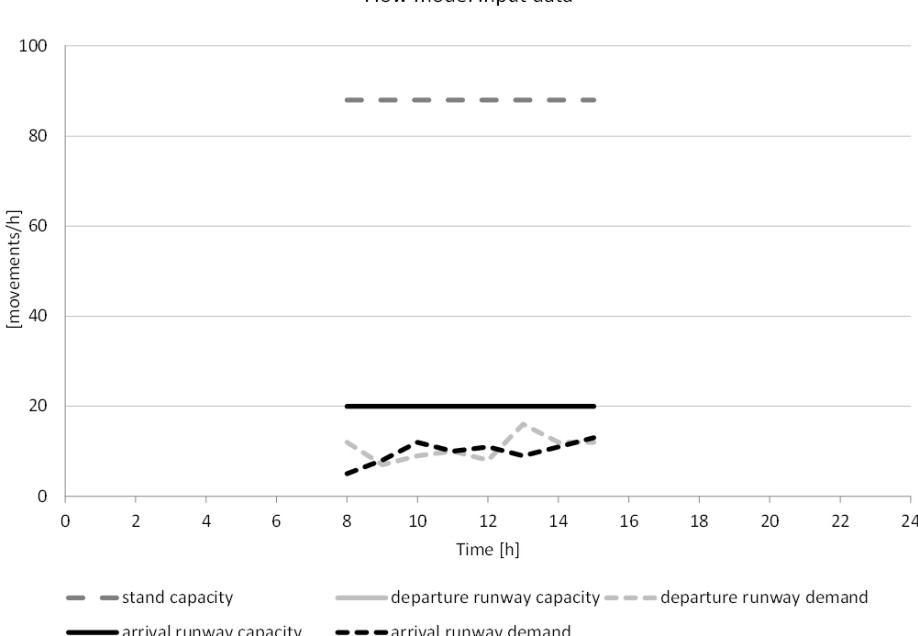

**Figure 6.** Extract of input data for the flow model.

The flow model provides primarily the traffic flow as an output. The traffic flow is the number of flights on an hourly basis that is handled by a resource. Moreover, the flow model calculates the number of delayed flights.

### 3.3.2. Event Model

The DLR milestone simulation, as the representative for event models, requires a minimum data set of the flight identifier, origin, destination, scheduled in-block times, and scheduled off-block times for each flight. Therefore, the provided flight plan data were used. Additionally, an assignment to the used runways, handling teams, and stand is necessary. The runway assignment was retrieved from the real flight trajectories as these provide the runway in use. The handling teams were assigned based on a simple scheduling algorithm because operational data were not available. The stand assignment was derived from the automatic assignment done by the motion model (cf., Section 3.3.3) to ensure comparability among the models.

Additionally, the milestone simulation requires process durations (e.g., for the final approach, taxi process, and ground handling). These data were taken from the previous Oslo project [17], where an operational analysis was conducted. Figure 7 shows an overview of the derived values.

The milestone simulation provides the actual times as defined within the Airport Collaborative Decision-Making (ACDM) Manual [9] as a result. Based on a comparison of scheduled and actual times, flight plan deviations can be calculated and the number of delayed flights can be derived.

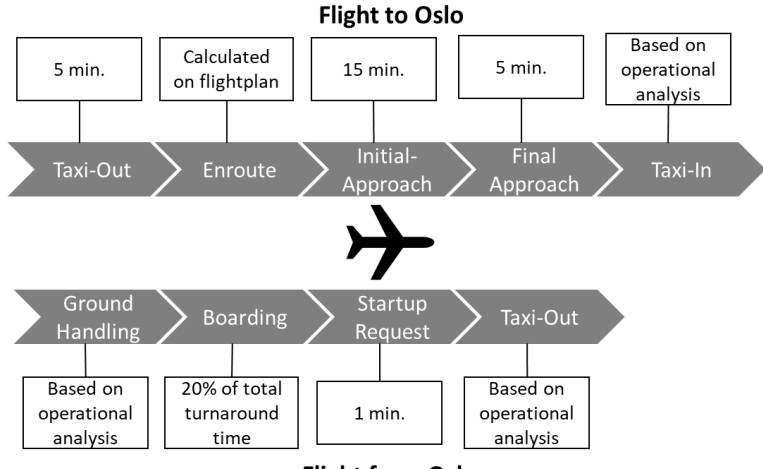

**Figure 7.** Summary of process durations for the DLR milestone simulation.

### 3.3.3. Motion Model

In terms of AirTOP, the simulation setup consists of the ground layout of Oslo airport and the related TMA. Both parts are modeled pursuant to the Norwegian AIP [25]. The ground layout comprises the runway system, taxiways, aircraft parking stands, and de-icing platforms. Regarding the TMA, the setup includes the flight procedures for departures and arrivals in detail until the first en route waypoint (departures) or from the last en route waypoint (arrivals). Additionally, the flights fly directly to their destination from the first en route waypoint (origin, Oslo) or fly directly from their origin to the last en route waypoint (destination, Oslo). Furthermore, speed and altitude restrictions can be applied and a wake turbulence separation can be set up [12]. AirTOP takes into account operational limitations such as snow removal and de-icing, which are described in Section 3.2.2. The traffic data for the flight plan were obtained through the DDR2 provided by EUROCONTROL (cf., Section 3.2.1). In each case, the created flight plans cover the aircraft movements of the entire selected day at Oslo airport. Additionally, the flight plans are linked, which means that a turnaround is considered.

Regarding the flight rules, the specifications of ICAO Doc 4444 [31] were respected in order to establish sufficient separation between the aircraft. If there were specific regulations concerning Oslo Airport, the Norwegian AIP was considered. The aircraft performance model is based on information from EUROCONTROL's base of aircraft data (version 3) [32], which means that the aircraft's motion could be simulated.

### 3.4. Evaluation Criteria

The simulation models were compared regarding accuracy and calculation speed (cf., Section 3.1). A high degree of accuracy allows the operators within airport management to improve their decisions as they have a better awareness of the future situation. A high calculation speed enables the conduction of a large set of scenarios as well as a fast transition from a situation assessment to a decision and the performance of an action [33]. For both dimensions, appropriate comparison indicators, so-called KPIs, need to be defined.

Accuracy has to be considered in detail. Airport management is an opportunity to react to crises and disruptions in operations. If flights become delayed, appropriate actions need to be considered. Therefore, it is necessary to understand where potential bottlenecks will occur and how many flights will be delayed. As such, the accuracy of the simulation model can be quantified by measuring the number of delayed flights. A flight is considered to be delayed when it is more than 15 min behind its schedule [34]. The closer the number of delayed flights calculated by the model is to the real number of delayed flights, the more accurate the model.

The number of delayed flights was calculated for each resource (arrival runway, stands, ground handling, and departure runway) so that the bottleneck could be located. The resources were subdivided into related phases (landing, in-block, takeoff, and off-block). KPI 1 is therefore the number of delayed flights per phase. It should be noted that negative delays (flights that arrive before their scheduled time) were ignored within this indicator. The number of delayed flights is sufficient to assess the impact of a certain event.

Nevertheless, from a decision-making point of view, different solutions need to be compared. In this case, the delayed flights might be misleading as the number of flights does not provide information on the level of delay. As a consequence, the average deviation per flight in comparison to the real average deviation per flight was used to additionally assess the models' accuracy. The deviation of a flight was defined as the difference between the actual time and the scheduled flight plan time within the landing, in-block, takeoff, and off-block phases. KPI 2 is therefore the average deviation per flight considering each phase.

The calculation time, as the third indicator, is a broadly used concept, but requires a detailed definition to provide valid conclusions. In airport management, time is a critical resource due to the large set of data elements, possible actions, and participating parties within the decision-making process [35]. Calculating the impact of a certain action manually is in principle possible as only basic knowledge of mathematics is necessary. The impact of a runway closure for instance can be calculated by shifting the first impacted flight by the duration of the closure and deriving the times for all following flights out of the maximum of the earliest runway time and the separation to the preceding flight. As a large number of flights and the effects on the resources (e.g., the stand usage, turnaround, and aircraft rotation) need to be considered, a manual calculation is not performable within a reasonable amount of time [17]. This is even more the case if multiple cases need to be evaluated. As such, the calculation time should be reduced by the what-if tool to a minimum so that operators have more time to make their decisions. The calculation time was measured as the time from the start of the simulation until the end of the simulation, which corresponds to the computation time. Additional time to present and understand the provided data was not considered within the indicator as this is a graphical design issue and outside the scope of this study. KPI 3 is therefore the average calculation time, which was derived after 50 replications.

It was assumed that models that simulate the airport in more detail (continuous rather than discrete models) gain a higher accuracy but require more calculation time.

## 4. Results

In this section, the results of each model's simulations are presented, focusing on the three KPIs that were specified in Section 3.4. Since we simulated two real days of operation (cf., Section 3.1), real data exists and can be used for comparison. Unfortunately, no real data were available for the in-block phase because the trajectory contained in the SO6 files ends at the aerodrome reference point.

### 4.1. KPI 1—Number of Delayed Flights

The analysis of the nominal scenario provided the following numbers of delayed flights. In reality, two arrivals and one departure were delayed within the corresponding phases. The flow model showed no delayed flights in any phase, whereas the event model and the motion model calculated in each case more than five delayed flights for the off-block and takeoff phases. Table 5 lists the complete values regarding the number of delayed flights per phase for the nominal scenario.

**Table 5.** KPI 1: Number of delayed flights per phase (nominal scenario).

|  | **Real Data** | **Flow Model** | **Event Model** | **Motion Model** |
|---|---|---|---|---|
| Landing | 2 | 0 | 2 | 0 |
| In-Block | not determinable | 0 | 2 | 0 |
| Off-Block | 1 | 0 | 7 | 6 |
| Takeoff | 1 | 0 | 7 | 6 |

In comparison to the real data, it can be seen that the off-block and takeoff phases have more delayed flights than the landing and in-block phases within the event model and the motion model. The real data show a different effect. Here, the off-block and takeoff phases have fewer delayed flights. If we look at the three different simulation models, we see that the motion model has fewer delayed flights than the event model within each phase but more delayed flights than the flow model in the case of the off-block and takeoff phases. The simulation results' discrepancy from the real data is 0% (event model) and −1.19% (flow model/motion model) for the in-block and landing phases and −0.58% (flow model), 3.51% (event model), and 2.92% (motion model) for the off-block and takeoff phases regarding the number of delayed flights per phase in the nominal scenario. In summary, the maximum discrepancy of the flow model occurs within the landing and in-block phases with −1.19%, whereas the event and the motion models show the highest discrepancy compared with the real data within the off-block and takeoff phases (3.51% and 2.92%, respectively).

The analysis of the non-nominal scenario provided the following numbers of delayed flights. In reality, 37 flights within the landing phase, 44 flights within the off-block phase, and 48 flights within the takeoff phase were delayed. The flow model showed no delayed flights within the landing and off-block phases, just two delayed flights within the in-block phase, and 56 delayed flights within the takeoff phase. The event and the motion models calculated more than eight delayed flights for each phase. Table 6 lists the complete values regarding the number of delayed flights per phase for the non-nominal scenario.

**Table 6.** KPI 1: Number of delayed flights per phase (non-nominal scenario).

|  | **Real Data** | **Flow Model** | **Event Model** | **Motion Model** |
|---|---|---|---|---|
| Landing | 37 | 0 | 18 | 16 |
| In-Block | not determinable | 2 | 18 | 14 |
| Off-Block | 44 | 0 | 9 | 13 |
| Takeoff | 48 | 56 | 26 | 38 |

It is evident that more flights were delayed in the non-nominal scenario than in the nominal scenario due to the influence of snowy weather. It can be observed that the simulation models have fewer delayed flights per phase than the real data except for one value. The exception is in the case of the takeoff phase, where only the flow model calculated more delayed flights. There is not an obvious trend apparent among the prediction tools as to which model reproduces the real data most precisely. Within the landing phase, the flow model shows a discrepancy compared with the real data of −22.2%, the event model shows a discrepancy of −11.4%, and the motion model shows a discrepancy of −12.6%. In the case of the in-block phase, a comparison to the real data was not possible, but here the flow model shows the fewest delayed flights and the event model the most. Regarding the off-block phase, the flow model has a discrepancy compared with the real data of −25.7%, the event model has a discrepancy of −20.5%, and the motion model has a discrepancy of −18.1%. Lastly, within the takeoff phase, the flow model has more delayed flights than the real data with a discrepancy of 4.68%, whereas the event model deviates by −12.9% and the motion model by −5.85% from the real data. Taken together, regarding KPI 1, the maximum discrepancy of each simulation model compared with the real data occurs within

the off-block phase. The flow model shows a discrepancy of −25.7%, the event model shows a discrepancy of −20.5%, and the motion model shows a discrepancy of −18.1%.

### 4.2. KPI 2—Average Deviation per Flight

Turning to KPI 2, the results regarding the nominal scenario show the following average deviation per flight and phase. Unfortunately, the flow model provides no results for KPI 2 because of its reduced complexity regarding the calculation. As mentioned in Sections 2.2 and 3.3.1, the flow model shows capacity constraints when comparing flow values with capacity values for different phases. Specific delays for each flight are not within the scope of the calculation. In reality, each phase shows a negative average deviation per flight with the largest deviation in the case of the landing phase. Consequently, each simulation model has a larger average deviation per flight than the real data for each phase. Figure 8 shows the results regarding the average deviation per flight and phase for the nominal scenario.

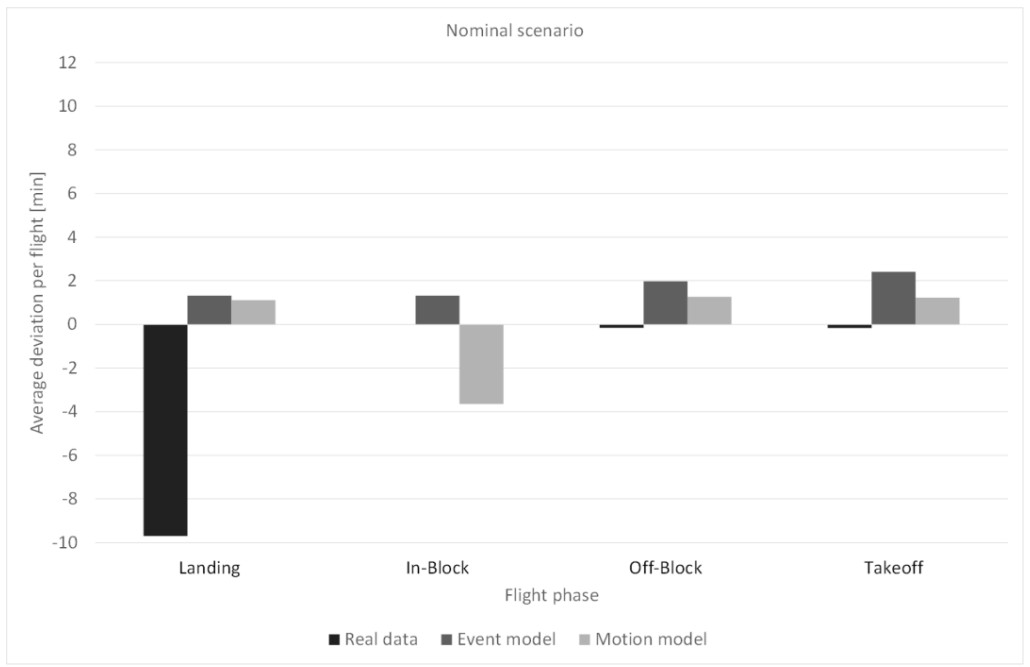

**Figure 8.** KPI 2: Average deviation per flight and phase (nominal scenario).

If we compare the two simulation models, we can see that the event model only has a positive deviation in each phase, whereas the motion model shows a negative deviation within the in-block phase. The largest discrepancy between the simulation models and the real data was determined for the landing phase with a difference of 11 min (event model) and 10.8 min (motion model). The discrepancy between the simulation models and the real data was much lower for the off-block and takeoff phases. Within the off-block phase, the difference was approximately 2.1 min for the event model and 1.4 min for the motion model. In the case of the takeoff phase, the event model deviates from the real data by approximately 2.6 min and the motion model deviates from the real data by 1.4 min. Regarding the nominal scenario, the motion model is the prediction tool that is closest to the real data in each phase. The event model always has a larger deviation than the motion model.

Looking at the non-nominal scenario, the results of KPI 2 show a major difference to the nominal scenario because the event model and the motion model have a smaller average deviation per flight than the real data for each phase. The largest discrepancy between the real data and the simulation models occurs within the off-block phase. Figure 9 shows the results regarding the average deviation per flight and phase for the non-nominal scenario.

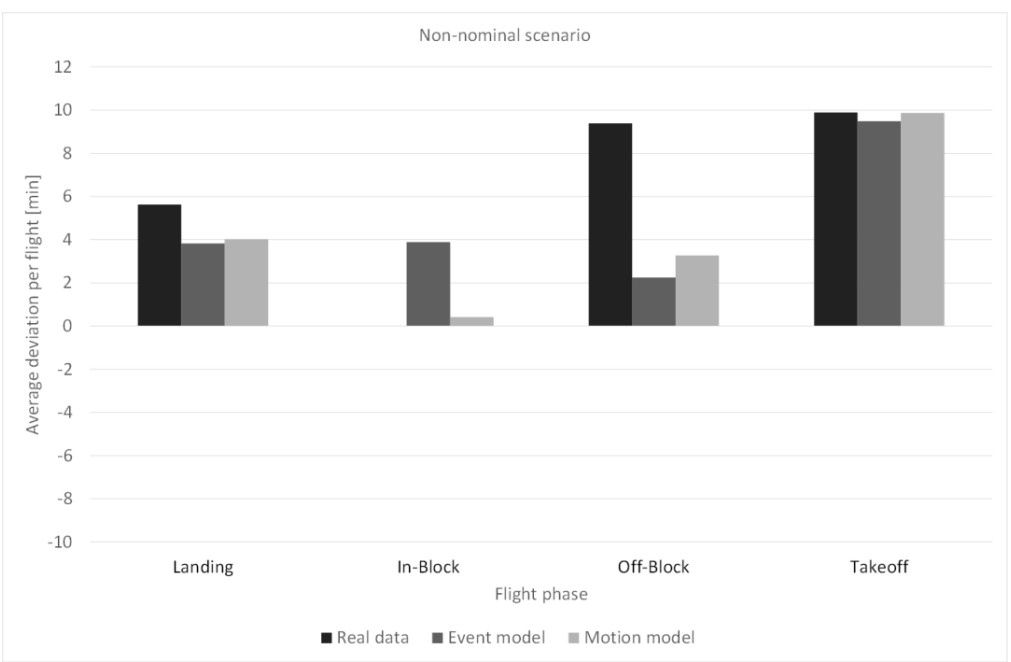

**Figure 9.** KPI 2: Average deviation per flight and phase (non-nominal scenario).

The event model and the motion model show similar trends of deviation values though the deviation is the opposite within the in-block phase. The difference to the real data within the landing phase is approximately 1.8 min for the event model and 1.6 min for the motion model. The largest difference between the real data and the simulation models is present within the off-block phase. Here, the event model deviates from the real data by approximately 7.1 min and the motion model deviates by 6.1 min. The takeoff phase shows the smallest discrepancy between the real data and the simulation models. The discrepancy is approximately 0.4 min for the event model and 0.02 min for the motion model. If we compare the simulation models, we see that the event model and the motion model have the largest difference to one another within the in-block phase (3.5 min). The diagram in Figure 9 shows that the results of the motion model are closest to the real data. This is consistent with the nominal scenario, where the motion model also delivered the results that are closest to the real data. Therefore, we can conclude that the motion model has the smallest discrepancy compared with the real data in the case of KPI 2.

*4.3. KPI 3—Calculation Time*

Turning now to KPI 3, the results regarding the average calculation time of each simulation model are presented in Figure 10. As mentioned in Section 3.4, the average calculation time was determined after 50 replications and comprises only the duration from the start of the simulation to the end of the simulation. It can be seen that there is a difference between the three prediction tools. Regarding the nominal scenario, the flow model has the shortest average calculation time (0.01 s), the event model has an average calculation time (1.18 s), and the motion model has the longest average calculation time (81.36 s). In relative terms, the flow model takes 0.012% of the motion model's calculation time to compute the results and the event model's calculation time is 1.45% of the motion model's calculation time.

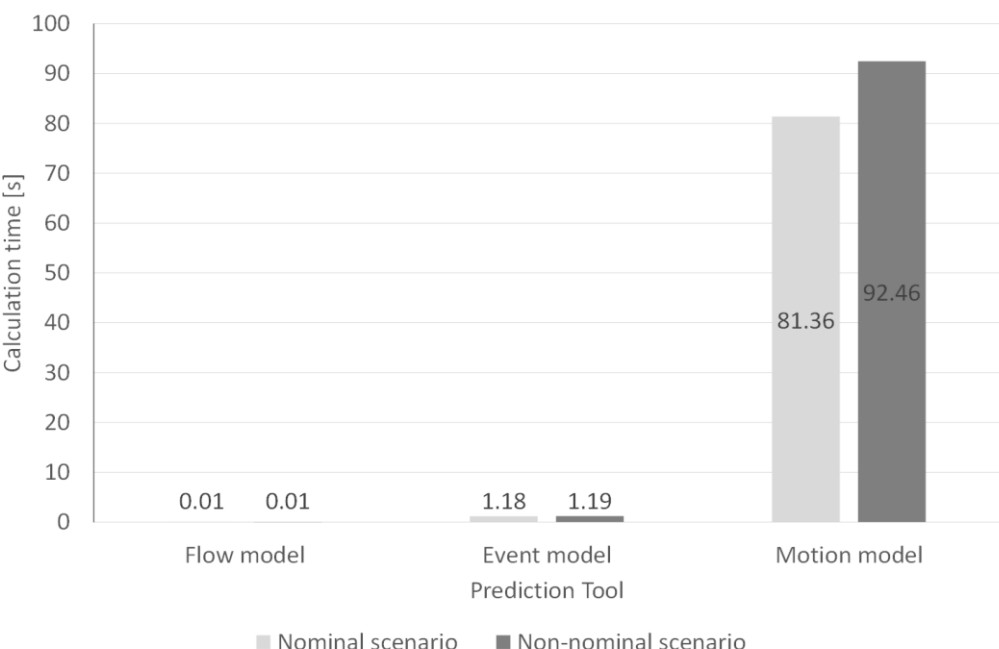

**Figure 10.** KPI 3: Average calculation time of each simulation model.

In the case of the non-nominal scenario, the flow model underlines the findings of the nominal scenario with an average calculation time of 0.01 s. Again, this is the shortest average calculation time among the simulation models. The event model takes 1.19 s of computation to calculate its results, whereas the motion model has once again the longest average calculation time (92.46 s). In relative terms, the flow model takes 0.011% of the motion model's calculation time to compute the results and the event model's calculation time is 1.29% of the motion model's calculation time. Generally speaking, no difference was found between the nominal scenario's calculation time and the non-nominal scenario's calculation time regarding the flow model. The calculation time of the event model and the motion model was always less in the case of the nominal scenario compared with the non-nominal scenario. Taking into consideration both scenarios, we can conclude that the flow model has the shortest average calculation time among the simulation models.

## 5. Discussion

In this section, we discuss the simulations' results and give a critical analysis regarding the assessment of the prediction tools.

### 5.1. KPIs

The results of KPI 1 regarding the nominal scenario show that there is only a slight discrepancy between the real data and the flow model/motion model for the landing phase. A larger discrepancy was noted for the off-block and takeoff phases. A possible explanation for this difference is that flights have a negative delay in the real data, which means that their real off-block time was earlier than their scheduled off-block time. Early arrivals are not considered by all simulation models since they depend on factors beyond the particular airport (e.g., the situation at the departure airport or en route weather conditions). Therefore, the largest discrepancy between the real data and the results of the event model and the motion model appears for the off-block and takeoff phases. When looking at the non-nominal scenario, the primary cause of the discrepancy for the landing phase was the occurrence of an Air Traffic Flow Management (ATFM) delay within the real data. It can be noted that 80 out of 167 arrivals were affected by ATFM regulations on this day. No prediction tool considers such regulative actions, since ATFM regulations are issued due to capacity constraints within the network. As en route sectors are not within the scope of the simulation tools, these effects could not be considered. One can observe from

Table 6 that there were no delayed flights within the landing phase in the case of the flow model because this prediction tool prioritizes arrival flights when single-runway operations are in use. Consequently, this leads to a large number of delayed flights within the takeoff phase. The delayed flights within the in-block phase are a consequence of occupied aircraft stands because the flow model considers capacity values for aircraft stands (cf., Figure 6). Exceeding the capacity values leads to delayed flights. With regard to the off-block phase, the flow model has a specific handling capacity (cf., Figure 6). If the demand exceeds the capacity, aircraft have to wait and will become delayed. This is a different approach compared with the event model and the motion model, where delays are caused by the landing delay plus the taxiing delay (in-block) as well as the rotational delay (off-block). There is no evident trend apparent with KPI 1 to finally assess the three different prediction tools, which is why we have to take a look at KPI 2.

Regarding KPI2 and the nominal scenario, the large negative deviation in the real data for the landing phase is very likely a result of the flight time planning. In reality, the scheduled flight times are calculated with additions to cover minor contingencies, whereas the scheduled simulative flight times do not have such additions. As was mentioned for KPI 1, the prediction tools cannot have negative deviations within the off-block phase. The results of KPI 2 show that the motion model is the only simulation model with a negative deviation in the case of the in-block phase. This result could be explained by the calculation of the in-block deviation. According to the Norwegian AIP, a taxiing time of 10 min has to be considered for flight planning of the scheduled arrival time [25], but the simulation results reveal that none of the 168 arrivals had a taxiing time equal to or greater than 10 min. Therefore, most of the flights arrived before their scheduled in-block time at the respective aircraft stands. Looking at the non-nominal scenario, the foremost cause of the discrepancy between the real data and the simulation models for the landing phase was the ATFM delay. The larger landing deviation in the real data then leads to a higher average off-block deviation because of the aircraft turnaround. Additionally, departing flights are also affected by ATFM delays, but this affected just 4 out of 171 flights. As was mentioned for the nominal scenario, the discrepancy between the event model and the motion model within the in-block phase can be explained by the calculation of the in-block deviation for the motion model. Taken together, we can derive from KPI 2 that the motion model delivered the results that came closest to the real data. The flow model showed an inability to compute results for this KPI, thus indicating a disadvantage in terms of accuracy.

The results of KPI 3 are in line with the expectations because, on the one hand, the motion model provides the most detailed forecast (cf., Section 2.2), thus leading to a higher calculation time. On the other hand, the flow model has the lowest accuracy but the fastest calculation time. The event model lies between the flow model and the motion model. This matches the findings because its accuracy lies between the two other simulation models. The calculation time needs to be regarded in reference to the available overall decision time because it is a subprocess within the decision process. Thus, the calculation time needs to be lower than the decision time. Additionally, the lower the calculation time, the more what-if cases can be calculated, which offers the possibility of assessing unlikely weather situations as well as a larger set of solutions. Previous human-in-the-loop experiments considered a decision time of 45 min [35]. The results of KPI 3 show that each simulation model has a calculation time that falls inside this timeframe. The flow model enables approximately 120 times as many runs as the event model and 8136 to 9246 times as many runs as the motion model. The event model can calculate 69 to 78 times as many runs as the motion model. On the basis of KPI 3, no simulation model can be regarded as unsuitable for a what-if tool, so the choice rests with the user and is dependent on the available calculation time within the overall decision time.

### 5.2. Summary

To sum up, the results show that no prediction tool outclasses the others in terms of accuracy and calculation time, although the flow model could not provide results for

KPI 2. Therefore, it would be necessary to define thresholds for the KPIs in the case of determining an acceptable model. The acceptable accuracy of a prediction tool is strongly dependent on the respective airport. The prediction tool may be less accurate if more free resources are accessible at the airport. In terms of an acceptable calculation time, this KPI is strongly dependent on the available decision time. If the stakeholders have more time for the decision process, higher calculation times are tolerable. This leads to the following conclusion. The selection of a prediction tool has to be made in dependence of the airport and the decision processes. The presented results show the advantages and disadvantages of each tool and, therefore, provide a first quantification by which a selection can happen. Besides accuracy and calculation time, the effort or expertise required to prepare and set up the simulation models should be considered as well. If a data base is available with existing capacity values, the effort required to use the flow model is low. In this case, and given the assumption that the average deviation is not necessary as an output, it is a suitable tool for what-if purposes. In contrast, simulation models such as the motion model require much more expertise to use. Here, an expert would be necessary for the use of this tool. If such an expert exists, a motion model would be suitable for what-if purposes.

## 6. Conclusions

The fundamental research question stated in the introduction was to determine which simulation model is most suitable for what-if purposes. The literature review discovered that a large number of airport simulation models exist, but a comparison of the four basic model types (cf., Section 2.1) in terms of what-if has not been conducted thus far. In this study, we investigated three different simulation model types by means of two days of operations (one day without the influence of weather effects on the airport and one day where snowfall affected an airport). These scenarios were replicated in each simulation model and afterwards simulated. To analyze the results, we defined three KPIs that enable a comparison with the real data and among the simulation models. These KPIs are the number of delayed flights per phase, the average deviation per flight and phase, and the average calculation time of the simulation models. The results of KPI 1 show no evident trend as to which simulation model came closest to the real data. Regarding KPI 2, it became clear that the motion model had the smallest discrepancy compared with the real data, whereas the flow model could not provide results for this KPI. In case of KPI 3, the flow model demonstrated its advantage in calculation time compared with the event and motion models. In general, the results suggest that, as expected, calculation time and accuracy are opposed. The model with the highest accuracy has the longest calculation time and vice versa. A consequence of this is that the simulation model has to be chosen regarding the circumstances at the airport. The flow model is suitable for airports where capacity values are known and where the what-if tool is exclusively used for quick decisions by the operational staff. In the case of accuracy and calculation time, the event model represents a trade-off between the flow model and the motion model. In our opinion, it is appropriate for airports where the durations of the operational processes and operational dependencies are known. Additionally, the airport should occasionally have simulative expertise available for the set-up and adjustment of the model. The motion model is an appropriate what-if tool for airports that have simulative expertise continuously available in the form of human simulation experts. This is necessary, because the set-up and operation of the tool requires much more effort and experience compared with the two other simulation models.

The present findings may help with the application of a what-if tool for airport management decisions. Future work should focus on quantification of the state model as well as evaluation with other operational scenarios. Future work should focus on the acceptance and application of the considered simulation models by the proposed users since they are not simulation experts. This may be a suitable additional KPI that supports the identification of the simulation model that is best placed to act as a prediction engine for what-if systems. A constraint of our research is the focus on Oslo airport. It would be possible to simulate various airports and analyze whether the results are different to the

findings of the present paper. Moreover, advancing automation regarding the calculation time and event transmission is a vital issue for future research. As an example, the required simulative expertise as well as preparation and calculation times can be reduced by digital automation. In particular, the motion model has a broad range of interfaces to communicate with the tool. This enables the possibility of implementing an automated interface that inputs events, such as weather effects, into the simulation in the form of occupancy times, runway closure times, and closed aircraft stands. A possible option would be automated processing of a METAR. The design and development of a self-learning system could be possible processes in the future.

**Author Contributions:** Conceptualization, O.P. and S.S.-M.; methodology, O.P., S.S.-M. and S.L.; software, O.P. and S.S.-M.; validation, O.P., S.S.-M. and S.L.; formal analysis, O.P.; investigation, O.P. and S.S.-M.; writing—original draft preparation, O.P.; writing—review and editing, S.S.-M. and S.L.; visualization, O.P., S.S.-M. and S.L. All authors have read and agreed to the published version of the manuscript.

**Funding:** This research received no external funding.

**Institutional Review Board Statement:** Not applicable.

**Informed Consent Statement:** Not applicable.

**Data Availability Statement:** Data are available from the corresponding author (Oliver Pohling). Access to the Demand Data Repository has to be granted by EUROCONTROL (https://www.eurocontrol.int/ddr).

**Acknowledgments:** Map data are copyrighted by OpenStreetMap contributors and available at https://www.openstreetmap.org.

**Conflicts of Interest:** The authors declare no conflict of interest.

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
