# Peer review of "Looking into the Crystal Ball—How Automated Fast-Time Simulation Can Support Probabilistic Airport Management Decisions"

_aerospace, doi:10.3390/aerospace9070389_

Round 1

Reviewer 1 Report

Manuscript ID: Aerospace-1778090

Title: Looking into the crystal ball – How automated fast-time simulation can support probabilistic airport management decisions

Dear Authors and Editors,

Thank you for inviting me to serve as one of the manuscript's reviewers. This study evaluated different types of prediction engines for airport management what-if systems by comparing them in terms of accuracy and calculation speed. The authors created three key performance indicators that can be compared to actual data and simulation models. These key performance indicators are the number of delayed flights each phase, the average variance per flight and phase, and the average simulation model calculation time. These findings are crucial because they enable stakeholders to better comprehend and regulate a what-if tool for airport management decisions.

The issue merits investigation and is within the scientific scope of the Journal. All editorial requirements have been met.  some of my constructive comments have been provided in the attachment:

In conclusion, I recommend acceptance with minor revision.

Best regards,

Author Response

Dear reviewer,

Thank you very much for your review report and your constructive feedback. Please see the attachment for our detailed response to your report. 

Kind regards

Reviewer 2 Report

The title and abstract reflect the content of the paper well.

The methodology is appropriately described, and the document sections are adequately structured.

In my opinion, it is ready for publication in the face of an improvement in the conclusions, adding some critical discussion of the results related to the Oslo airport, further research steps, and avoiding any quotations.

Author Response

Dear reviewer,

Thank you very much for your review report and your constructive feedback. Please see below our response to your remarks:

We added the limitation to Oslo airport and empasized additional possible future research steps to improve the findings. Futhermore, we removed the last two sentences with the quotation from the present paper.

Kind regards

Reviewer 3 Report

Dear Authors,

Your paper is interesting and deals with an important topic for airport management.

Please see below some comments regarding your paper:

1) Introduction you must make your contribution clearer, in what way your paper is different from others?

2) Section 3 . Table 5 please present it in a better way; remember that the reader must understand the table without reading the text.

3) Section 4. Maybe presenting a Summary Table with your results should be interesting to readers. Also if is possible to present the cost of each model, should be interesting.

Kind Regards,

Author Response

Dear reviewer,

Thank you very much for your review report and your constructive feedback. Please see the attachment for our detailed response to your remarks.

Kind regards
